# Pair production of charged IDM scalars at high energy CLIC

**Jan Klamka**⋆

Faculty of Physics, University of Warsaw

⋆ Jan.Klamka@fuw.edu.pl

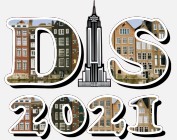 *Proceedings for the XXVIII International Workshop on Deep-Inelastic Scattering and Related Subjects, Stony Brook University, New York, USA, 12-16 April 2021*

## Abstract

The Compact Linear Collider (CLIC) was proposed as the next energy-frontier infrastructure at CERN, to study $e^+e^-$ collisions at three centre-of-mass energy stages: 380 GeV, 1.5 TeV and 3 TeV. The main goal of its high-energy stages is to search for the new physics beyond the Standard Model (SM). The Inert Doublet Model (IDM) is one of the simplest SM extensions and introduces four new scalar particles: $H^\pm$, A and H; the lightest, H, is stable and hence a natural dark matter (DM) candidate. A set of benchmark points is considered, which are consistent with current theoretical and experimental constraints and promise detectable signals at future colliders.

Prospects for observing pair-production of the IDM scalars at CLIC were previously studied using signatures with two leptons in the final state. In the current study, discovery reach for the IDM charged scalar pair-production is considered for the semi-leptonic final state at the two high-energy CLIC stages. Full simulation analysis, based on the current CLIC detector model, is presented for five selected IDM scenarios. Results are then extended to the larger set of benchmarks using the DELPHES fast simulation framework. The CLIC detector model for DELPHES has been modified to take pile-up contribution from the beam-induced $\gamma\gamma$ interactions into account, which is crucial for the presented analysis. Results of the study indicate that heavy, charged IDM scalars can be discovered at CLIC for most of the proposed benchmark scenarios, with very high statistical significance.

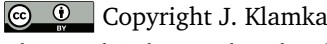

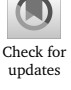
## 1 Introduction

The Compact Linear Collider (CLIC) [1] is an $e^+e^-$ collider proposed as the next energy frontier machine at CERN. At the first running stage, at $\sqrt{s} = 380$ GeV, it will allow many precision measurements, in particular Higgs boson and top quark studies. CLIC also offers access to many new physics phenomena at its two high-energy running stages, at 1.5 TeV and 3 TeV,

providing a discovery reach complementary to that of the LHC [2]. This talk presents the CLIC sensitivity to production of new heavy particles predicted by the Inert Doublet Model (IDM).

The IDM [3] is one of the simplest extensions of the SM. In addition to the SM Higgs doublet, a new "inert" doublet is introduced with four scalar fields: $H^{\pm}$, A and H. The lightest of these, H, is stable thanks to the $Z_2$ symmetry, which makes it a natural DM candidate. After electroweak symmetry breaking and fixing of the SM parameters, five free parameters are left: three masses of the IDM scalars and two coupling constants.

A total of 23 scenarios are considered, with high IDM scalar masses selected from the two sets of bechmark points in the model parameter space proposed in [4]. They respect all current theoretical and experimental constraints, as well as cover all interesting areas of the parameter space in the context of future lepton collider studies. Each benchmark corresponds to different values of the inert scalar masses and couplings, resulting in a range of production cross sections (depending almost entirely on the scalar masses, as the influence of couplings is marginal). Only the pair production of IDM particles is possible in lepton colliders, with the neutral and charged scalar pair production as the two dominant channels:

$$
\begin{aligned}
e^+e^- &\rightarrow \text{ H A}, \\
e^+e^- &\rightarrow \text{ H}^+\text{H}^-.
\end{aligned}
$$

The scalar A further decays into Z and H, while $H^{\pm}$ predominantly into $W^{\pm}$ and H. It is important to note that, depending on the scalar mass splittings in both channels, the gauge bosons produced in these decays can be either virtual or real.

## 2 Strategy and the analysis

The search for the IDM scalars at CLIC was previously considered in the leptonic channel (with both W and Z decaying into lepton pairs) in a generator level study [5]. However, the sensitivity turned out to be limited to cross sections (for scalar production and decay in the considered final state) of about 1 fb and the discovery was possible only up to scalar masses $m_A + m_H \sim 550\,\text{GeV}$ and $m_{H^{\pm}} \sim 500\,\text{GeV}$, with many scenarios beyond the discovery reach. To extend the discovery reach, the semi-leptonic final state is considered in the analysis described here, with one W decaying into jets and another leptonically. This signature is only possible for the charged IDM scalar pair production, but offers almost one order of magnitude higher cross sections than the leptonic signature.

For the event generation WHIZARD 2.7.0 [6] was used, with CLIC beam spectra taken into account and −80% electron beam polarisation assumed. The strategy was to use full simulation of detector response based on GEANT4 [7] and DD4HEP [8] packages to investigate CLIC sensitivity for five selected scenarios. All 23 benchmark points were then considered using the fast simulation DELPHES [9] package, with dedicated CLIC detector (CLICdet) cards [10], to consider a wide range of scenarios, while still taking into account the detector response.

In the event selection, applied for the signal and background discrimination, reconstruction output was required to include an electron or muon and a pair of jets corresponding to the expected final state signature. A simple cut-based preselection was also imposed, based on the distributions of kinematic variables describing the system. Figure 1 shows the distributions of mass and energy of a dijet system, obtained using fast simulation for the two signal scenarios (HP17 and BP23) and the SM background. There is a significant difference visible between the two benchmarks. This is because the mass splitting $m_{H^{\pm}} - m_H$ is small in the HP17 scenario, and the produced W boson is highly virtual, while in BP23 it is produced on shell.

Events that passed the preselection have been considered in the event classification procedure. To optimise the signal event selection, Boosted Decision Trees (BDTs) implemented

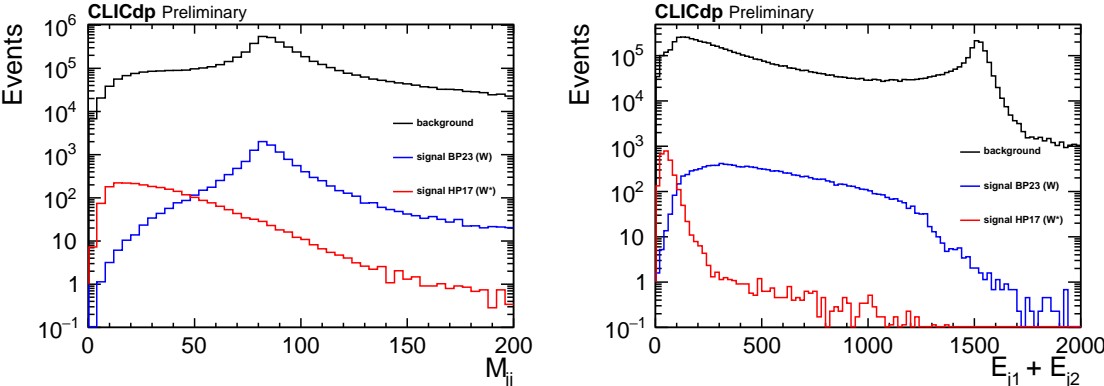

Figure 1: Histograms of the mass (left) and the energy (right) of a dijet system, obtained using fast simulation, for $\sqrt{s} = 3$ TeV. The red line denotes HP17, and the blue one BP23. Black histogram is the sum of all SM background channels. Normalization is the number of events expected in the real experiment.

in the TMVA toolkit [11] were used. They were trained separately for the two datasets: one composed of scenarios with off-shell W production and the second one containing remaining signal samples with on-shell W bosons.

## 2.1 Influence of the overlay events

Because of the high beam intensity and bunch collision rate, the beam-induced backgrounds are an integral part of the CLIC physics. The most important, from the point of view of event reconstruction, is $\gamma\gamma$ interactions producing soft hadrons, the so-called "overlay events". This is crucial in this analysis for scenarios with small scalar mass difference: highly virtual W bosons in signal events decay into low-energy jets and leptons, the reconstruction of which is strongly influenced by the $\gamma\gamma \to$ hadrons processes.

Standard treatment of the overlay events is to apply cuts on the time stamps of reconstructed Particle Flow Objects (PFOs), allowing this background to be reduced (this is reflected in the full simulation). Unfortunately, the timing cuts are not implemented in the CLICdet model for DELPHES. Therefore, an additional generator-level selection was applied to the $\gamma\gamma \to$ hadrons samples, before overlying them on the signal and background samples, to imitate the timing cuts on the PFO level. The impact of this procedure on di-jet and single jet mass distributions is presented in Fig. 2 for the HP17 scenario (with small $m_{H^\pm} - m_H$). Compared are respective distributions produced using full simulation and DELPHES, with and without the overlay contribution. One can see the clear improvement in the description of the full simulation results by the fast simulation after including the $\gamma\gamma$ interactions.

## 3 Results

Using the BDT classification results, the expected statistical significance of deviations observed from the SM background was calculated for each benchmark scenario. The significance resulting from the analysis based on the full simulation is shown in Fig. 3 (left), as a function of $2m_{H^\pm}$, for 3 TeV CLIC running stage and the five considered IDM scenarios. For two of them (with the lowest scalar mass values) a real W boson is produced and a virtual W boson for the other three. The smallest $m_{H^\pm}$ corresponds to the BP23 scenario, while the third largest to the HP17 (which also has the smallest $m_{H^\pm} - m_H \approx 10$ GeV of the five shown scenarios).

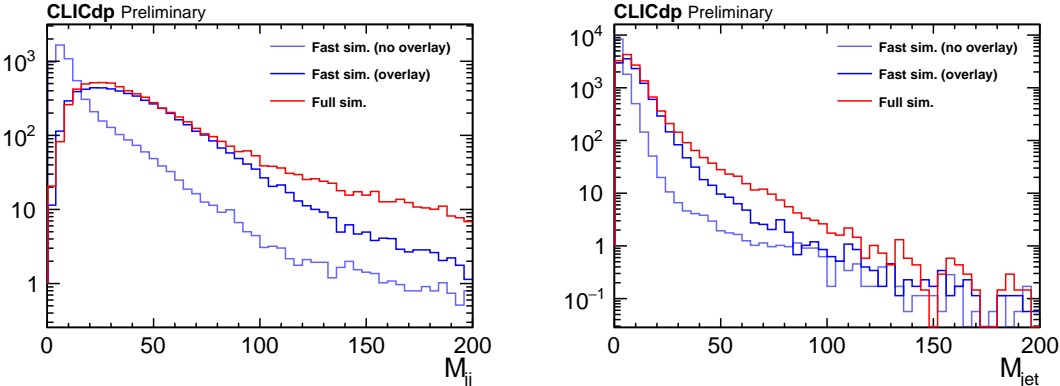

Figure 2: Histograms of the mass of a di-jet system (left) and a single jet (right), obtained using different simulation methods for signal HP17 at $\sqrt{s} = 3$ TeV. The red line shows the full simulation, azure the fast simulation and blue the fast simulation with overlay contribution.

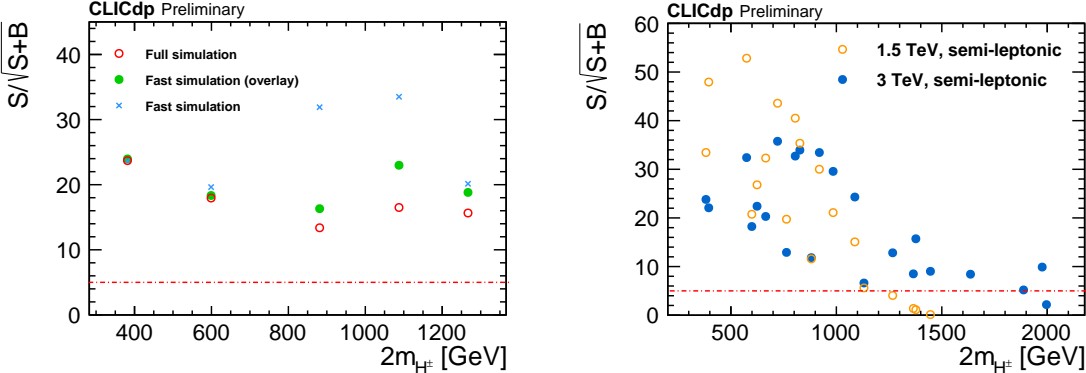

Figure 3: Statistical significance expected in the study, as a function of $2m_{H^\pm}$. The comparison of results obtained with different simulation methods for 3 TeV CLIC (left) and the full set of the results from fast simulation with overlay background included (right). The red dotted line shows $5\sigma$ threshold. See text for more details.

The results are compared with those obtained using fast simulation, both with and without the $\gamma\gamma \rightarrow$ hadrons contribution. For a proper validation of the DELPHES results, as well as to avoid uncontrolled bias in such a small datasets, the BDTs were trained on each scenario separately both for the fast and the full simulations. It is visible that after including the overlay background the results obtained with fast simulation are much closer to those based on the full simulation. It confirms that this technique leads to realistic predictions.

The resulting significance as a function of $2m_{H^\pm}$, for all 23 benchmark scenarios considered in the study, is presented in Fig. 3 (right) for two CLIC high-energy stages. The results are based on fast simulation with the overlay events included and BDTs trained simultaneously for all signal scenarios, as described in Sec. 2. The presented results show that the charged IDM scalars can be observed at CLIC, with high statistical significance reaching $50\sigma$, for masses up to about 1 TeV.

# 4 Conclusion

The feasibility of detecting heavy charged IDM scalar pair production at CLIC was studied. Detector response was modeled for five selected signal scenarios with full simulation based on GEANT4 and the study was then extended to all 23 considered benchmark scenarios using DELPHES. The $\gamma\gamma \rightarrow$ hadrons overlay background was included in the fast simulation, as its influence on the reconstruction of the signal events cannot be neglected in case of small scalar mass splittings. We conclude that the observation of charged IDM scalar pair-production, with their masses up to 1 TeV, is possible almost for all considered benchmark scenarios.

# Acknowledgements

The work was carried out in the framework of the CLIC detector and physics (CLICdp) collaboration. We thank collaboration members for fruitful discussions, valuable comments and suggestions. This work benefited from services provided by the ILC Virtual Organisation, supported by the national resource providers of the EGI Federation. This research was done using resources provided by the Open Science Grid, which is supported by the National Science Foundation and the U.S. Department of Energy's Office of Science.

The work was partially supported by the National Science Centre (Poland) under OPUS research project no. 2017/25/B/ST2/00496 (2018-2021).

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
