# Peer review of "Pair production of charged IDM scalars at high energy CLIC"

_SciPost Physics Proceedings, doi:SciPost Phys. Proc. 8, 097 (2022)_

## Round 1 · Referee Report · Anonymous (Referee 1) · 2022-2-28

Report

The paper, "Pair production of charged IDM scalars at high energy CLIC", describes the Inert Doublet Model, a natural extension to the standard model, in which 4 beyond the standard model particles are predicted. Specifically a study is performed for the discovery reach of two high energy CLIC stages in which a semi-leptonic decay of the pair produced IDM particles is detected. The author succinctly covers the theoretical justification for the search, and explains the signal and background that would need to be understood in the analysis of the actual data measured by the future CLIC.

There are a few minor comments that should be considered before the paper is published. It would be preferable to have a slightly more extended discussion of the difference between the 23 benchmark scenarios. While there is a brief statement that the scenarios cover all the interesting parameter space proposed by future lepton colliders, it would round out the discussion contained in this paper to expand on the differences between the scenarios. Figure 3 was quite useful in this regard, as it showed the different 2m_H^{\pm} mass values in the benchmark scenarios. In a similar vein, a little more explanation of the HP17 and BP23 scenarios might improve clarity in this document.

The suggested added discussion does not need to be comprehensive, but adding a few sentences to improve clarity would be appreciated.

I believe this paper satisfies the requirements of this journal once the above comments are taken into consideration.

---

## Round 2 · Author Response

Resubmitted the updated version, with the recommendation from the referee taken into account. Added a small comment in Sec. 1, which clarifies the differences between the benchmark points. Also, a comment in Sec. 3 was added to expand on the BP23 and HP17 scenarios and explain to which points in Fig. 3 they correspond.

---

## Round 2 · List of Changes

• Sec. 1 (Introduction): a comment on what is the difference between the benchmark points
  • Sec. 3 (Results): reference of the BP23 and HP17 scenarios to the results shown in Fig. 3 (left)

---

## Editorial Decision

published